# Coarse-Grained Molecular Dynamics Simulations of Lipid Nanodroplets and Endosomal Membranes: Focusing on the Fusion Mechanisms

**DOI:** 10.3390/ijms262411960

**Published:** 2025-12-11

**Authors:** Yeon Ju Go, Erkhembayar Jadamba, Hyunjin Shin

**Affiliations:** MOGAM Institute for Biomedical Research (MIBR), Seoul 06730, Republic of Korea; yeonju.go@mogam.re.kr

**Keywords:** coarse grained molecular dynamics simulations, ionizable lipid, endosomal membrane, endosomal escape, lipid nanodroplet, lipid nanoparticle, membrane fusion mechanism

## Abstract

Lipid nanoparticles (LNPs) have received significant attention as effective RNA carriers in RNA-based therapeutics and vaccines. Particularly, ionizable lipids (ILs) of LNPs play a crucial role in endosomal escape and lipid-mediated RNA delivery owing to their pH-dependent molecular characteristics. Therefore, it is essential to enhance the endosomal escape efficiency of ILs, which is primary bottleneck in the successful cytoplasmic delivery of RNA. However, the molecular-level understanding of the roles and dynamics of ILs during the endosomal escape stage remains unclear. To elucidate this, we utilized coarse-grained (CG) molecular dynamics (MD) simulations. In this simulation, we designed lipid nanodroplets (LNDs) containing D-Lin-MC3-DMA (MC3) and ALC-0315, which have proven effective as LNPs in RNA-based therapeutics and vaccines, respectively, while accounting for the pH environments of early and late endosomes. Also, we formulated lipid bilayers reflecting the composition of early and late endosomal membranes to investigate the fusion process between LNDs and endosomal membranes. Our findings reveal that, irrespective of endosomal membrane composition and LNP types, ILs are the first lipids to enter the endosomal membrane during the fusion, and the flip-flop process of ILs from the inner leaflet to the outer leaflet of the endosomal membrane is a critical step for LNP endosomal escape. More specifically, we observed that protonated ILs predominantly participate in the flip-flop process, while many deprotonated ILs remain clustered and disordered within the intermediate layer of the endosomal membrane. Furthermore, we found that the extent of IL flip-flop varies with the cholesterol content of the endosomal membrane. Additionally, under identical pH conditions, MC3-containing LNDs exhibited a more active IL flip-flop process toward the outer leaflet than ALC-0315-containing LNDs. This observation supports experimental findings that MC3-containing LNPs manifest higher endosomal escape efficiency than ALC-0315-containing LNPs in mRNA delivery studies. The mechanistic insights into the endosomal escape mechanism demonstrated by our simulations could aid in the development of effective ILs.

## 1. Introduction

RNA-based therapeutics have demonstrated great potential not only as treatments for various diseases such as cancer, neurodegenerative disorders, and metabolic syndromes through the suppression of pathogenic gene expression but also as effective vaccines [1,2,3,4,5]. Despite their therapeutic promise, the clinical application of RNA therapeutics remains challenging due to their large molecular size, negative charge, and hydrophilicity, which hinder cellular uptake, as well as their susceptibility to degradation by nucleases [6,7]. To overcome these biological barriers, a range of delivery vectors has been developed, among which lipid nanoparticles (LNPs) have emerged as the most clinically successful RNA delivery platform [3,8]. To date, three LNP formulated RNA medicines have been approved by the Food and Drug Administration (FDA). These include Onpattro, a small interfering RNA therapy for transthyretin mediated amyloidosis, and two messenger RNA (mRNA) vaccines against COVID-19 developed by Moderna and Pfizer BioNTech [9,10,11]. A common feature of these LNP formulations is the incorporation of ionizable lipids (ILs) as a key component. ILs remain neutral at physiological pH, thereby minimizing toxicity, but become positively charged in acidic environments, facilitating RNA encapsulation and promoting endosomal escape [12,13,14,15,16]. Endosomal escape is widely regarded as the major bottleneck in RNA delivery, with studies estimating that less than 5% of internalized RNA successfully reaches the cytosol [17,18]. Therefore, understanding the endosomal escape properties of ILs is essential for the development of more efficient LNPs, and extensive research is currently underway to address this challenge [19,20]. Notably, an experimental study demonstrated that LNPs incorporating ALC-0315 [9], the IL used in Pfizer and BioNTech’s mRNA vaccines, undergo distinct structural transitions along the endosomal and lysosomal maturation pathway at the mesophase level [19]. However, due to limitations in spatial and temporal resolution, experimental approaches have difficulty capturing the dynamic interactions between LNPs and endosomal membranes during the escape process. As such, in silico approaches, particularly molecular dynamics (MD) simulations [21,22], offer a powerful means to overcome these limitations and provide molecular level insight into the mechanisms of endosomal escape. MD simulations can capture the structural changes and stability of lipids during membrane fusion at the molecular level over time [23,24]. In particular, for simulating large systems involving interactions between LNPs and membranes over microsecond timescales, it is more practical to use coarse grained (CG) MD simulations, which are tens to hundreds of times faster than all atom (AA) MD simulations [25,26,27,28].

Thus, we performed hundreds of microseconds of CG MD simulations using Martini 3 force fields [27,29] together with newly developed Martini 3-compatible CG parameters for ILs to investigate the fusion process of LNPs into the endosomal membrane (Figure 1). For this simulation, we utilized MC3, an IL effective in intravenously delivered LNP-formulated RNA therapeutics, and ALC-0315, which performs well in LNP-formulated RNA vaccines administered intramuscularly [30,31]. The Martini force field model is widely used for observing dynamics in various biological systems due to its acceptable reliability and has been extensively used in membrane fusion studies [24,25,26,28]. Notably, the Martini CG model has been employed in studies aimed at uncovering the molecular mechanisms of lipoplexed DNA endosomal escape, which is closely related to our interest [24]. However, previous computational studies did not clarify how protonated and deprotonated ILs differentially contribute to membrane fusion, leaving the roles of ILs that depend on their protonation state largely unresolved [23,24]. In contrast, our study shows that protonation-dependent IL behavior, especially the flip-flop of protonated ILs, is a key mechanistic factor that contributes to the fusion required for endosomal escape.

Unlike previous computational studies, we designed several lipid nanodroplets (LNDs) as LNP-mimicking lipid mixtures with different protonated states depending on pH and created endosomal membranes that reflect the lipid compositions of early and late endosomes (Figure 1). This could facilitate the identification of more realistic LNP endosomal escape mechanism. On the other hand, we further simplified the endosomal membranes while effectively retaining their anionic characteristics (Figure 1), because it is challenging to perfectly reflect the composition of endosomal membranes, which varies with the endosomal stage or cell type.

By analyzing the CG MD trajectories of multiple combined LND and endosomal membrane systems (Figure 1), we uncovered the roles of ILs at the onset of LND–endosomal membrane fusion, as well as the distinctive dynamics of protonated and deprotonated ILs during fusion. Additionally, by observing IL dynamics under variations in endosomal membrane composition, we inferred the role of cholesterol in modulating IL dynamics during fusion. Furthermore, analysis of the clustered size and morphology of LNDs after fusion revealed the clustering characteristics of protonated and deprotonated ILs. Examination of the second-rank order parameter of protonated and deprotonated ILs further identified the ordering characteristics of their lipid tails. Our simulation results thus elucidated the distinctive behaviors of protonated and deprotonated ILs when MC3-containing and ALC-0315-containing LNDs fuse with endosomal membranes under different pH conditions and membrane compositions. These insights can deepen our understanding of the fusion process not only for LNPs containing MC3 and ALC-0315 but also for other IL-LNPs, thereby facilitating the design of more efficient LNPs.

## 2. Results and Discussion

### 2.1. Designing LNDs with ILs and Endosomal Membrane

To prevent payloads such as RNA from degrading in highly acidic environments, LNPs must escape before the endosome matures into the lysosome [12]. LNPs are considered feasible for endosomal escape via membrane fusion not only because they have cell-like structures but also due to the interaction between positively charged lipids within the LNP and the anionic characteristics of the endosomal membrane [32]. However, the molecular mechanism governing the endosomal escape of LNPs remains poorly understood. Therefore, we focused on the membrane fusion process to gain a deeper understanding of this mechanism at the molecular level.

To explore the endosomal escape mechanism of LNPs with ILs, we first designed LNDs using LNP-mimicking lipid mixtures with ILs in a CG representation, and separately designed endosomal membranes. We then employed CG MD simulations to uncover the molecular-level dynamics occurring during the fusion of LNDs with endosomal membranes. We constructed the LNDs using two distinct ILs: MC3 and ALC-0315. The lipid compositions were designed to mimic those of the Pfizer-BioNTech mRNA vaccine [33] and a benchmark MC3-LNP-siRNA formulation [16]. The ILs, 1,2-distearoyl-sn-glycero-3-phosphocholine (DSPC), and cholesterol (CHOL) were used to construct the LNDs (Figure 2).

Additionally, to aid in understanding a realistic LNP endosomal escape mechanism, we designed LNDs and endosomal membranes that reflect critical characteristics such as pH changes and compositional changes in the endosomal membrane during endosomal maturation along the endosomal and lysosomal maturation pathway. To account for changes in the fraction of protonated ILs within LNDs due to the pH variations inside the endosome, we designed 11 types of LNDs for both ALC-0315-containing LNDs and MC3-containing LNDs, incrementing the fraction of protonated ILs from 0% to 100% in increments of 10%. Detailed methods of LNDs design are described in Section 3.1.

Next, we designed the endosomal membranes by mimicking the in vivo composition [34], varying the ratios of 1-palmitoyl-2-oleoyl-sn-glycero-3-phosphocholine (POPC), 1,2-dioleoyl-sn-glycero-3-phosphoethanolamine (DOPE), and CHOL to create early endosomal membrane (EEM) and late endosomal membrane (LEM). Conversely, reflecting the complete composition of endosomal membranes is challenging due to the lack of precise knowledge about the asymmetry in composition between the inner and outer leaflets of the lipid bilayer and its variation with endosomal stage or cell type. Thus, we designed endosomal membranes that reflect the anionic characteristics while simplifying the membrane into a form commonly used in simulation studies [24]. This involved using 80 mol% POPC as the major phospholipid and 20 mol% of the anionic lipid 1-palmitoyl-2-oleoyl-*sn*-glycero-3-phospho-L-serine (POPS) to simplify the endosomal membrane. We define this simplified endosomal membrane as the simple lipid bilayer (SLB) and will refer to this throughout the paper. The endosomal membranes used in this study are all in the form of lipid bilayers. We chose this form because surrounding endosomes with LNDs would be computationally costly, given their large size, and our goal is to understand the interactions between LNDs and immediately adjacent parts of the endosomal membrane upon fusion. Additionally, other computational studies investigating the interactions between bio-membranes and LNDs also designed cellular membranes as lipid bilayers [23]. The detailed method for designing endosomal membranes is described in Section 3.2.

### 2.2. Designing LND–Endosomal Membrane Complex and CG MD Simulations for Membrane Fusion

In the previous section, we created a total of 22 LNDs by combining both ALC-0315-containing and MC3-containing LNDs with three types of endosomal membranes, namely EEM, LEM, and SLB, thereby producing the LND–endosomal membrane systems. Using the known pH ranges of early endosomes (6.8–5.9) [35] and late endosomes (5.5–5.0) [36], along with the apparent pK_a_ values of ALC-0315 (6.09) [37] and MC3 (6.44) [38], we calculated the protonated ratios of ALC-0315 and MC3 within LNDs in each endosomal environment using the Henderson–Hasselbalch equation. Consequently, the protonated ratio of ALC-0315 was found to be 16.3–60.6% in early endosomes and 79.5–92.5% in late endosomes. Similarly, MC3 exhibited a protonated ratio of 30.4–77.6% in early endosomes and 89.7–96.5% in late endosomes. For the protonated cases of LNDs, designed in 10% increments, LNDs with a protonated ratio of ALC-0315 of 0–70% were combined with EEM, and those with a ratio of 70–100% were coupled with LEM, resulting in 12 LND–endosomal membrane complexes. For the case where ALC-0315 had a 70% protonated ratio, the uncertainty surrounding its clear assignment to EEM or LEM led to its inclusion in both designs. Similarly, LNDs with a protonated ratio of MC3 of 0–80% were coupled with EEM, while 90–100% were associated with LEM, resulting in 11 complexes. Additionally, combinations of ALC-0315 and MC3 protonated ratios of 0–100% with SLB resulted in a total of 22 LND–endosomal membrane complexes. On all 45 complexes, we performed 135 unbiased CG MD simulations for 2000 ns in three replicas. Using the resulting simulation trajectories, we analyzed the structural changes in lipids during the fusion process between LNDs and endosomal membranes from multiple perspectives (Figure 3). This analysis allowed us to uncover the characteristics and roles of lipids and elucidate the fusion process between LNDs and endosomal membranes. Although these simulations provided meaningful results, the conventional CG MD approach used in this study employs fixed protonation states for ILs. Consequently, it cannot capture the protonation changes that may occur due to the local lipid environment [39], such as during the fusion of LNDs with endosomal membranes. To address this limitation, constant-pH MD simulations or other adaptive protonation approaches, which allow protonation states to respond to the surrounding environment, could offer a more accurate description of lipid dynamics during the fusion process [40,41]. Despite their high computational cost and the challenges associated with achieving convergence in lipid systems with pH-dependent structural transitions and force field sensitivity [42], such methods may reveal additional mechanistic details in future studies.

### 2.3. The Onset of Fusion Between LNDs and Endosomal Membranes

Based on the well-known role of ILs in promoting RNA endosomal escape [13,14,15,16], we hypothesized that ILs initiate fusion by strongly binding to the endosomal membrane as LNPs begin to interact with it. Our first goal was to illustrate this from a structural perspective at the molecular level. To achieve this, we analyzed 135 CG MD simulation trajectories, each lasting 2000 ns, from the previous section. This analysis aimed to capture the moment when LNDs start to fuse with the endosomal membrane and to identify the lipids that first penetrate the endosomal membrane during this process. To pinpoint the onset of fusion, we calculated the average z-coordinate of PO4 CG beads from POPC molecules in the upper leaflet of the lipid bilayer corresponding to the endosomal membrane. We then determined the difference (ΔZ) between the lowest z-coordinate among CG beads within the LNDs (h_L_) and the calculated upper leaflet’s average z-coordinate (h_u_) (Figure 4A). Fusion was identified as starting when ΔZ shifted from positive to negative and maintained a negative value for at least 1 ns thereafter (Figure 4B). The onset of fusion varied from 1.4 to 1458 ns across different systems (Figure 4C). In 117 out of 135 systems (86.7%), the residue in the LND showing the smallest ΔZ was identified as IL. In the remaining 18 systems (13.3%), 14 systems had IL as the residue with the second smallest ΔZ just before fusion occurred, while in the last 4 systems, IL was the third smallest ΔZ residue. Although the IL exhibited the second or third smallest ΔZ among the 18 systems, visual inspection using VMD [43] confirmed that the residues with the smallest ΔZ began fusing with the endosomal membrane nearly simultaneously. Consequently, our simulation results indicate that in nearly all instances, IL is the first component to enter the membrane when LNDs begin to fuse with the endosomal membrane, playing a significant role in the fusion process. This suggests that the pH-dependent charge of ILs within LNDs might facilitate strong electrostatic interactions with endosomal membranes, thereby overcoming the hydration repulsion barrier, which is theorized to prevent spontaneous lipid membrane fusion [44]. Unlike natural bio-membrane components, the inclusion of ILs in LNDs appears to aid this process.

### 2.4. Flip-Flop Process of ILs in Endosomal Membrane

Once LNDs have fused with the endosomal membrane, lipids can exhibit dynamics such as rotation, lateral diffusion, and flip-flop [45]. For LNDs to successfully escape the endosomal membrane, fusion must occur not only with the inner leaflet but also with the outer leaflet of the endosomal membrane. Thus, we hypothesized that flip-flopping, which involves lipid exchange between the two leaflets of the bilayer (Figure 5), would be crucial. Furthermore, given that ILs play a pivotal role in LNP endosomal escape, we considered that the flip-flop process of ILs is particularly important for the LNDs’ endosomal escape mechanism.

To address this, we analyzed 135 sets of 2000 ns CG MD trajectories to calculate the population of ILs located in the outer leaflet of the membrane following the fusion of LNDs with the endosomal membrane. For precise analysis, snapshots of the LND–endosomal membrane complex were taken at 200 ps intervals from the CG MD trajectories, specifically within the 1500–2000 ns range where complete fusion was determined for all systems. The position of ILs in the endosomal membrane was based on the location of the terminal beads of the head groups of ALC-0315 and MC3, which were used as ILs. The terminal CG beads for the head groups of ALC-0315 and MC3 are OH and NC3, respectively (Figure 6A). These beads were defined as belonging to the inner leaflet if their minimum distance to any PO4 bead of a POPC molecule forming the surface of the inner leaflet was less than or equal to 1.2 nm. Similarly, they were defined as belonging to the outer leaflet if their minimum distance to any PO4 bead of a POPC molecule forming the surface of the outer leaflet was less than or equal to 1.2 nm. Beads that did not belong to either the inner or outer leaflets were identified as being in the midplane (Figure 6B). Using the population of head group beads located on the outer leaflet surface, we were able to determine the flip-flop rate of ILs. This is supported by two observations. First, ILs containing head groups positioned on either the inner or outer leaflet surfaces were aligned nearly parallel to the normal vector of the lipid bilayer (Figure 6C). Second, when the movement of ILs with head groups eventually found on the outer leaflet surface was tracked over the 0–2000 ns CG MD trajectories using VMD [43], ILs were visually observed to flip-flop only once from the inner leaflet to the outer leaflet. Additionally, ILs with head groups located in the midplane were confirmed to exist in a disordered state (Figure 6D).

Utilizing the previously described method, we calculated the migration rates of protonated and deprotonated ILs to the outer leaflet and midplane across 45 distinct LND–endosomal membrane systems. With three replicas for each system, we computed the average and standard deviation to determine these rates. To further assess the robustness of our simulations, we examined the migration rates obtained from the three replicas and confirmed that they exhibit consistent trends across systems (Appendix A). In addition, time-evolution plots of fusion onset (Δ*Z*) and IL migration fractions for a representative system are provided in Appendix A, illustrating the temporal behavior leading to the post-fusion regime. The 45 systems were designed based on the type of ILs, the percentage of protonated ILs within LNDs, and the specific type of endosomal membrane. In systems where the proportion of protonated ALC-0315 varied from 0 to 100% in increments of 10%, and paired with either EEMs or LEMs, we found that a significant fraction of protonated ALC-0315, ranging from 0.29 to 0.44, migrated to the outer leaflet. This rate was markedly higher than that of deprotonated ALC-0315, which moved to the outer leaflet at a rate of 0.02 to 0.11 (Table 1). In contrast, deprotonated ALC-0315 predominantly migrated to the midplane, with rates ranging from 0.65 to 0.97, compared to 0.10 to 0.48 for protonated ALC-0315. Most deprotonated ALC-0315 remained centrally within the endosomal membrane, exhibiting minimal movement to the inner or outer leaflets (Table 1). These findings highlight that, regardless of the protonated state of LNDs, the flip-flop of ILs plays a crucial role in membrane fusion for LNP endosomal escape, with protonated ILs significantly contributing to the flip-flop process.

Similarly, in scenarios where the endosomal membrane is SLB, the dynamics resemble those observed with EEMs or LEMs. Protonated ALC-0315 demonstrates a high migration rate to the outer leaflet, ranging from 0.44 to 0.59, while showing a lower migration rate to the midplane, between 0.06 and 0.17. Conversely, deprotonated ALC-0315 exhibits a lower migration rate to the outer leaflet, ranging from 0.11 to 0.34, and a higher rate to the midplane, between 0.47 and 0.76 (Table 2). This indicates that, irrespective of the endosomal membrane’s composition, ILs actively undergo flip-flop, with protonated ILs playing a crucial role in this process.

However, when the endosomal membrane transitions from EEM or LEM to SLB, the behavior of protonated ALC-0315 shows a notable shift. The migration rate to the outer leaflet increases from 0.29–0.44 to 0.44–0.59 across all protonated cases (Figure 7A). Conversely, the rate of migration to the midplane decreases from 0.10–0.48 to 0.06–0.17 across all protonated instances (Figure 7B). For deprotonated ALC-0315, the migration rate to the outer leaflet rises from 0.02–0.11 to 0.11–0.34 in all protonated cases (Figure 7C). Simultaneously, the rate to the midplane declines from 0.65–0.97 to 0.47–0.76, exhibiting a similar trend to that seen with protonated ALC-0315 (Figure 7D).

These differences are likely attributed to variations in endosomal membrane composition. Specifically, SLB lacks cholesterol, whereas EEM contains approximately 36%, and LEM contains about 18%. As the cholesterol content in the membrane increases, the migration of both protonated and deprotonated ILs to the outer leaflet tends to decrease (Figure 7A,C), while their migration to the midplane increases (Figure 7B,D). Considering the established fact that cholesterol interacts with other lipids in the cellular membrane to promote clustering and enhance lipid lateral packing [46,47,48,49], it is inferred that increased cholesterol levels may inhibit the flip-flop of ILs and elevate their clustering in the midplane. Moreover, during the fusion of LNDs with the endosomal membrane, the cholesterol within LNDs may also influence the dynamics of ILs. Therefore, adjusting the composition of LNDs based on the endosomal membrane’s composition appears important. Although our endosomal models do not explicitly include sphingomyelin (SM) or bis(monoacylglycerol)phosphate (BMP), incorporating these lipids would likely reinforce the trends observed in our simulations across SLB, EEM, and LEM. SM is relatively enriched in EEM, where its strong interaction with cholesterol promotes lateral packing and reduces membrane fluidity [50,51]. In contrast, BMP is uniquely localized to the inner leaflet of late endosomes and lysosomes and is known to enhance the activity of lipid transfer proteins such as saposins and Niemann–Pick disease type C2, thereby facilitating cholesterol mobilization and extraction [50,51]. These properties are consistent with the lower cholesterol content and higher membrane fluidity observed in LEM. Therefore, even when SM and BMP are considered, the simulation-derived conclusion that reduced cholesterol content increases IL flip-flop would remain unchanged. Furthermore, BMP is a negatively charged, cone-shaped phospholipid restricted to the inner leaflet of late endosomes [52]. Owing to these structural and electrostatic features, BMP-rich membranes are thought to facilitate membrane fusion [53]. Consequently, in BMP-rich and highly protonated late endosomal environments, the flip-flop of protonated ILs would be expected to increase.

MC3-containing LNDs exhibited the same trends observed with ALC-0315-containing LNDs. Regardless of the protonated state of LNDs and the composition of the endosomal membrane, protonated MC3 had a higher migration rate to the membrane’s outer leaflet and a lower rate to the midplane. Conversely, deprotonated MC3 showed a lower migration rate to the outer leaflet and a higher rate to the midplane (Appendix A). Additionally, when the endosomal membrane transitions from EEM or LEM to SBL, both protonated and deprotonated MC3 generally increased their migration rate to the outer leaflet while decreasing their rate to the midplane (Figure 8). Specifically, for protonated MC3, the migration rate to the outer leaflet was 0.35–0.55 with EEM or LEM, and 0.31–0.58 with SLB, increasing in all protonated cases except the 10% protonated case. In contrast, the rate at which protonated MC3 occupied the midplane decreased from 0.07–0.29 with EEM or LEM to 0.05–0.06 with SLB across all protonated cases (Figure 8, Appendix A). Similarly, deprotonated MC3’s migration rate to the outer leaflet increased from 0.01–0.12 with EEMs or LEMs to 0.13–0.19 with SLB in all protonated cases. Conversely, the rate at which deprotonated MC3 occupied the midplane decreased from 0.79–0.97 with EEM or LEM to 0.71–0.75 with SLB across all protonated cases (Figure 8, Appendix A).

Namely, MC3-containing LNDs exhibit the same trends as those observed with ALC-0315-containing LNDs. However, it was noted that, under identical protonated conditions and even at the same pH levels, protonated MC3 tends to show a higher proportion in the outer leaflet compared to protonated ALC-0315 (Figure 9). Figure 9 presents the population ratios of protonated ALC-0315 and protonated MC3 in the outer leaflet of the endosomal membrane during the fusion of ALC-0315-containing and MC3-containing LNDs with EEM or LEM. These ratios are shown according to the protonated case and pH of the LNDs. Except for cases where the fraction of ILs in the LNDs is 80% or 100%, the number of protonated MC3 molecules in the outer leaflet was greater than that of ALC-0315 (Figure 9A). Additionally, by determining the pH corresponding to specific protonated cases for both ALC-0315-containing and MC3-containing LNDs (Section 3.5), and comparing LNDs with similar pH levels, it was found that the proportion of protonated MC3 migrating to the outer leaflet was higher than that of protonated ALC-0315 across all pH environments (Figure 9B). For deprotonated ILs, there is minimal difference in the ratios of deprotonated ALC-0315 and deprotonated MC3 migrating to the outer leaflet across all protonated cases of LNDs, with both exhibiting low migration rates (Appendix A). Thus, considering the overall IL population, the number of MC3 molecules reaching the outer leaflet is greater than that of ALC-0315.

We hypothesized that a higher number of ILs flopping to the outer leaflet would facilitate more efficient membrane fusion during endosomal escape for LNDs. Consequently, we anticipated that MC3-containing LNDs, which have a higher proportion of ILs in the outer leaflet, would exhibit greater endosomal escape efficiency compared to ALC-0315-containing LNDs. This aligns with experimental results showing that MC3-containing LNPs have higher endosomal escape efficiency than ALC-0315-containing LNPs in mRNA delivery studies, with MC3 achieving 5% cytosolic delivery compared to 4% for ALC-0315 [54]. Although the numerical difference is modest, endosomal escape is widely regarded as a major bottleneck in intracellular drug delivery, with typically fewer than 5% of nucleic acid cargo reaching the cytosol. Therefore, even a 1% difference can represent a biologically meaningful distinction. While our current study does not address membrane perturbation energy directly, future computational studies that map the free-energy landscape of lipid insertion or quantitatively estimate fusion rates could provide a deeper mechanistic understanding of LNP–membrane interactions and endosomal escape.

### 2.5. LNDs Clustering

In the previous section, we calculated the positions of ILs in the membrane following the fusion of LNDs with the endosomal membrane, revealing that deprotonated ILs predominantly reside in the midplane. However, the degree of clustering among these lipids remained unclear, leading us to conduct a clustering analysis on both protonated and deprotonated ILs. To achieve this, radial distribution functions (RDFs) were calculated for each type of IL. The analysis was performed using snapshots taken at 200 ps intervals from the 135 CG MD trajectories of LND–endosomal membrane complexes, specifically within the 1500–2000 ns range. The RDFs were computed using only the NC3 CG beads of the ILs’ head groups, utilizing GROMACS 2024 [55]. The results revealed that deprotonated ILs tend to cluster closely in all systems, whereas protonated ILs are generally more dispersed. Figure 10 depicts the RDFs for protonated and deprotonated ALC-0315 in a system where LNDs and EEM are complexed with 50% protonated ALC-0315. Despite having equal numbers of deprotonated and protonated ALC-0315 in the membrane, deprotonated ALC-0315 showed an RDF peak at 0.52 nm with a maximum value of 43.6, while protonated ALC-0315 had a peak at 0.86 nm with a maximum value of 8.9. This indicates that deprotonated ALC-0315 clusters closely, whereas protonated ALC-0315 does not. The same trend was observed across the remaining 134 systems, even when deprotonated ALC-0315 was markedly less abundant (Appendix A).

We further analyzed the maintenance of cluster size in LNDs during fusion with the endosomal membrane, focusing on the presence of protonated and deprotonated ILs within the largest clusters. This analysis utilized homemade code integrated with MDAanalysis [56,57], assuming that molecules are clustered if the minimum distance between them is 1.2 nm or less. Analyzing the maximum clustering size from snapshots extracted at 200 ps intervals from 135 sets of 2000 ns CG MD trajectories, we observed that initially, the number of lipids forming LNDs was around 100, but decreased after fusion with the endosomal membrane (Figure 11). As the proportion of protonated ILs in LNDs increased, the maximum cluster size reduced. Notably, while the number of deprotonated ILs in the max cluster remained stable over time, the number of protonated ILs decreased significantly. For instance, LNDs with 10% protonated ALC-0315 contained 45 deprotonated ALC-0315, matching the count of protonated ALC-0315 in LNDs with 90% protonation. Over time, the average number of deprotonated ALC-0315 in the max cluster of LNDs with 10% protonation was 20, whereas the average number of protonated ALC-0315 in the max cluster of LNDs with 90% protonation was merely 5, indicating poor clustering maintenance by protonated ALC-0315 (Figure 11A,B). Similarly, in LNDs with 50% protonated ALC-0315, both protonated and deprotonated ALC-0315 initially numbered 25 each, but differed in retention within the max cluster after fusion with the endosomal membrane (Figure 11C). MC3-containing LNDs yielded similar results to those observed with ALC-0315-containing LNDs (Figure 11D–F). Based on previous findings that deprotonated ILs tend to cluster while protonated ILs disperse, and deprotonated ILs predominantly occupy the membrane’s midplane whereas protonated ILs are located near the surface, we conclude that deprotonated ILs help maintain LND shape by clustering in the midplane, while protonated ILs detach and spread near the membrane surface. Our simulation results are consistent with earlier studies using cryo-TEM, SAXS, NMR, and MD simulations, which reported that neutral ionizable aminolipids reside between the two membrane leaflets, while protonated lipids remain at the bilayer surface [58,59]. Furthermore, cryo-TEM analyses of Lipid-5-containing LNPs have revealed a single lamellar region surrounding an amorphous core [60], and MD simulations have captured the formation of amorphous droplets composed of neutral Lipid-5 within the hydrophobic core [61]. Together, these experimental and computational observations support our finding that deprotonated ILs preferentially cluster in the hydrophobic midplane region of the membrane.

### 2.6. Second-Rank Order Parameter of Lipid Tails

To assess the ordering of IL lipid tails during LND fusion with the endosomal membrane, we calculated the second-rank order parameter, P_2_, for IL tails. P_2_ is defined as ⟨(1/2)(3 cos^2^θ − 1)⟩, where θ is the angle between the vector formed by two beads of the lipid tails and the normal vector of the lipid bilayer. P_2_ value of 1 indicates that lipid tails are perfectly parallel to the bilayer’s normal vector, 0 indicates a completely disordered orientation, and negative values suggest that the tails are perpendicular to the bilayer’s normal vector. Using snapshots taken every 200 ps from the 135 CG MD trajectories within the 1500–2000 ns range, we calculated the bond order parameter for consecutive tail bonds and the tail order parameter for non-consecutive bead vectors of ALC-0315 and MC3 tails, combining these into a segmental order parameter. As illustrated in Figure 12, in a system where LNDs with 50% protonated ALC-0315 fuse with EEM, the segmental order parameters for beads composing the sn1 and sn2 tails of ALC-0315 are nearly identical due to their same chemical structure (Figure 12A). Across all segments, protonated ALC-0315 tails exhibited higher segmental order parameters than deprotonated tails. Notably, deprotonated ALC-0315 segments showed values close to 0, indicating a relatively disordered arrangement compared to protonated segments (Figure 12B). For segments T1–GL1 (T6–GL2), T2–T3 (T7–T8), T4–T5 (T9–T10), the segmental order parameters were 0.16 (0.16), 0.09 (0.09), and 0.09 (0.09), respectively (Figure 12B). When the endosomal membrane was SLB, segments T1–GL1 (T6–GL2), T2–T3 (T7–T8), T4–T5 (T9–T10) had order parameters of 0.25 (0.25), 0.12 (0.13), and 0.12 (0.12), respectively (Figure 12C), indicating lower values for ALC-0315 tails in EEM compared to SLB. Similar trends were observed for MC3 (Figure 12D), where protonated MC3 segments showed higher segmental order parameters than deprotonated segments, which had values close to 0, indicating a highly disordered orientation (Figure 12E,F). Additionally, when comparing hydrophilic head-adjacent segments CA1–CA2 (CB1–CB2), the segmental order parameter was 0.24 (0.24) in EEM and increased to 0.29 (0.29) in SLB, showing a 0.5 increase in SLB for MC3 tails (Figure 12E,F). Comparing protonated ALC-0315’s tail order parameter to protonated MC3’s, protonated ALC-0315’s T1–T3 (T6–T8) segments had order parameters of 0.15 (0.15) in EEM (Figure 12B) and 0.22 (0.22) in SLB (Figure 12C), whereas protonated MC3’s CA1–CA5 (CB1–CB5) segments were 0.24 (0.24) in EEM (Figure 12E) and 0.28 (0.28) in SLB (Figure 12F), indicating MC3’s tail order parameter was higher than that of ALC-0315.

Across all CG MD trajectories, we consistently found that protonated ILs have larger segmental order parameter values and are more ordered compared to deprotonated ILs, which tend to have values near 0, indicating a more disordered arrangement. Additionally, when the endosomal membrane composition is EEM or LEM, the segmental order parameters for ILs are lower than those observed in SLB. This suggests that in SLB, where the POPC ratio is higher and the cholesterol ratio is lower, IL tails achieve better packing and exhibit more order than in EEM or LEM. Furthermore, the tail order parameter for MC3 is greater than for ALC-0315, suggesting that MC3 achieves better lipid packing and order compared to ALC-0315. We also calculated the segmental order parameter for POPC, the most abundant lipid in the endosomal membrane, to assess differences in packing and order influenced by IL types. When LNDs with 50% protonated ILs fused with EEM, the segmental order parameter values for POPC tails, sn1 and sn2, were higher in the presence of MC3 compared to ALC-0315 (Figure 13A,B). This result was consistent when LNDs with 50% protonated ILs fused with SLB (Figure 13C). These results suggest that ALC-0315 contributes to membrane fusion by slightly increasing membrane disorder and fluidity compared to MC3. However, this difference is minor, and MC3 is known to have higher endosomal escape efficiency than ALC-0315 in in vitro studies [54]. Thus, the variations in disorder and fluidity between ALC-0315 and MC3 do not significantly impact the trend of MC3’s superior endosomal escape efficiency. Instead, these findings highlight the critical role of the flip-flop process in the endosomal escape mechanism. Additionally, regardless of the IL type used, the segmental order parameter for POPC’s saturated chain (sn1) was higher than for the unsaturated chain (sn2), indicating more regular packing in saturated chains (Figure 13B,C). This observation aligns with previously established knowledge [62].

## 3. Materials and Methods

### 3.1. LNDs Design

We designed two distinct types of LNDs utilizing ALC-0315 and MC3, which are ILs employed in FDA-approved LNP-formulated RNA medicines. All LNDs consisted of 100 lipids with a molar ratio of IL:CHOL:DSPC set to 50:40:10. In fact, the actual formulation for RNA vaccines and therapeutics uses a molar ratio of IL:CHOL:DSPC:PEGylated lipid at 50:38.5:10:1.5 [16,33]. However, due to the small proportion of PEGylated lipids potentially introducing artifacts, they were excluded, simplifying the molar ratio to IL:CHOL:DSPC at 50:40:10 for constructing LNDs. For each type of IL, we designed 11 types of LNDs by varying the proportion of protonated ILs from 0 to 50 in increments of 5. The initial coarse-grained structures of CHOL and DSPC were obtained using the INSert membrane tool (https://github.com/Tsjerk/Insane.git (accessed on 11 April 2024)) [29,63], and CG MD simulations were performed using the MARTINI 3 force field [27]. However, since the CG force field (CG FF) parameters for protonated and deprotonated ALC-0315 and MC3 are not available in the Martini version, we first performed AA MD simulations using known AA force fields (AA FF). Simultaneously, the CGbuilder tool [64] was employed to convert the AA structures of ILs into corresponding CG representations (Appendix A). Using exploratory CG FF parameters for ILs, we performed CG MD simulations. Subsequently, the structural information obtained from AA MD simulation trajectories was used as a reference to fit the structural data derived from CG MD simulation trajectories, allowing us to determine optimal CG FF parameters [65] (Appendix A). The initial structures and AA FF parameters for protonated and deprotonated ALC-0315 were based on Petra Čechová et al., 2024 [23], while those for protonated and deprotonated MC3 were obtained using CHARMM-GUI (https://www.charmm-gui.org (accessed on 21 September 2021)) [66,67,68,69]. Detailed procedures for obtaining the initial CG structures and CG FF parameters are provided in the Appendix A under the section “CG force field parameterization for ALC-0315 and MC3.” To strengthen the reliability and transferability of the newly developed CG parameters, we additionally performed a quantitative validation of the CG models against AA reference simulations. Specifically, we compared mass density, radius of gyration, and the root mean square deviation of RDFs (RMS Δg) between AA and CG simulations for both protonated and deprotonated ALC-0315 and MC3. All metrics showed AA–CG agreement with deviations within reasonable CG–AA accuracy ranges, and the detailed numerical values are provided in Appendix A. After obtaining the initial CG structures and CG FF parameters for each lipid, we used GROMACS [55] tools to randomly place the lipids in a 12 × 12 × 12 nm^3^ box and solvate them with standard MARTINI water particles. Then, we neutralized the system and achieved a physiological concentration of 0.15 M by adding Na^+^ and Cl^−^ ions. Subsequently, all systems underwent energy minimization using GROMACS 2024 [55] for refinement. The energy minimization process was terminated when the maximum force convergence criterion reached below 100 kJ mol^−1^ nm^−1^. Following this, CG MD simulations were conducted under NPT conditions for 1700 ns to achieve equilibration, using a time step of 20 fs.

### 3.2. Lipid Bilayer Design

To simulate the early and late endosome membranes, we mimicked the composition found in vivo [34]. The early endosome membrane was designed with a composition of POPC, DOPE, and Cholesterol in the ratios of 43%, 21%, and 36%, respectively. For the late endosome membrane, the ratios of POPC, DOPE, and CHOL were adjusted to 65%, 17%, and 18%, respectively. Although SM and BMP are also present in endosome lipid compositions found in vivo, they were not included in our models to maintain a simplified and representative lipid composition based primarily on POPC, DOPE, and CHOL. Additionally, we designed a simplified endosomal membrane model reflecting the anionic characteristics of endosomes with a composition of 80% POPC and 20% POPS. All lipid bilayer systems were constructed using the INSert membrane tool [29,63]. Each system was built within a simulation box of 12 × 12 × 10 nm^3^, with lipids solvated using standard MARTINI water beads. The early endosome membrane model comprises 192 POPC, 94 DOPE, and 162 CHOL molecules, along with 6676 MARTINI water particles. The late endosome membrane model includes 292 POPC, 76 DOPE, and 80 CHOL molecules, solvated with 6508 water particles. The simple lipid bilayer system consists of 360 POPC and 90 POPS molecules, together with 6311 MARTINI water particles. To neutralize the system, 0.15 M NaCl was added to all systems. Subsequently, the systems underwent minimization until the maximum force convergence was less than 10 kJ mol^−1^ nm^−1^. Finally, to relax the solvent and lipid bilayer, the systems were equilibrated for 2200 ns under NPT conditions with semi-isotropic pressure coupling using 20 fs integration time steps.

### 3.3. LND–Endosomal Membrane Complex Design

Based on the 22 LND types constructed in the previous section and the three types of endosomal membranes, a total of 45 LND–endosomal membrane complexes were generated by combining each LND with the appropriate endosomal environment as described in Section 2.2. The LND was manually positioned above the endosomal membrane using VMD [43] such that the minimum distance between the two components was approximately 1 nm. To avoid artifacts arising from periodic boundary conditions along the membrane normal, simulation boxes were constructed such that the minimum distance between the bottom of the complex and the edge of the box was 10 Å, and the minimum distance between the top of the complex and the upper box boundary was at least 100 Å, allowing the complex to be solvated in standard MARTINI water particles. Specifically, complexes with EEM were placed in a simulation box with dimensions of 10.7 × 10.7 × 25.0 nm^3^, those with LEM in 11.5 × 11.5 × 25.0 nm^3^, and those with SLB in 12.2 × 12.2 × 25.0 nm^3^. Each system was neutralized by adding NaCl at a concentration of 0.15 M. Energy minimization was performed using the steepest descent method [70] until the maximum force was below 100 kJ mol^−1^ nm^−1^. For each of the 45 systems, three independent replicas were generated, resulting in a total of 135 systems. All systems were simulated for 2000 ns under NPT conditions with isotropic pressure coupling, using a time step of 20 fs.

### 3.4. Simulation Protocol

All coarse-grained simulations were performed using GROMACS 2024 [55] with the Martini 3 force field [27], along with custom-developed CG parameters for the ILs. Periodic boundary conditions were applied in all three dimensions. The verlet cutoff scheme was used with a neighbor list update every 20 steps and a buffer tolerance of 0.005 nm. Non-bonded interactions were calculated using a 1.1 nm cutoff for both van der Waals and Coulombic interactions, employing potential-shift modifiers to ensure a smooth decay to zero at the cutoff distance. A relative dielectric constant of 15 was used to account for electrostatic screening in the Martini water model. The system temperature was maintained at 310 K using the velocity-rescaling thermostat [71] with a time constant of 1.0 ps. Pressure was controlled isotropically using the C-rescale barostat [72], with a reference pressure of 1 bar, a coupling time constant of 4.0 ps, and a compressibility of 4.5 × 10^−5^ bar^−1^.

### 3.5. Calculating Protonated Ratio and pH of LNDs

The pH environments of early and late endosomes are known to be 6.8–5.9 and 5.5–5.0, respectively. The apparent pK_a_ values for ALC-0315-containing LNPs and MC3-containing LNPs are 6.09 and 6.44, respectively. Using the Henderson–Hasselbalch equation, we calculated the range of protonated ratios for LNDs in each endosomal environment. The Henderson–Hasselbalch equation is defined as pH = pK_a_ + log([A^−^]/[HA]), where HA is the acid and A^−^ is the conjugate base; here, HA represents protonated IL and A^−^ represents deprotonated IL. Results indicated that the protonated ratio of ALC-0315 in LNDs ranged from 16.3% to 60.6% in early endosomes and from 79.5% to 92.5% in late endosomes. Similarly, MC3 in LNDs showed a protonated ratio of 30.4% to 77.6% in early endosomes and 89.7% to 96.5% in late endosomes. Furthermore, the pH values corresponding to each protonated case for ALC-0315-containing and MC3-containing LNPs were calculated using the Henderson–Hasselbalch equation (Table 3).

## 4. Conclusions

Developing ILs that enhance the endosomal escape of LNPs is crucial for the efficient delivery of RNA into the cytoplasm. However, the dynamics and roles of ILs in this process are poorly understood at the molecular level. To elucidate this, we employed CG MD simulations to examine IL dynamics during the membrane fusion process, a key strategy for LNP endosomal escape. We designed more realistic LNDs and endosomal membranes by considering the pH changes and compositional variations in the endosomal membrane as the endosome matures, and observed their fusion using CG MD simulations. The analysis of CG MD simulation trajectories revealed that ILs play a crucial role when LNDs begin to fuse with the endosomal membrane, suggesting that ILs interactions help overcome the kinetic barriers that impede spontaneous lipid membrane merging. Moreover, our analysis of ILs’ positions on the endosomal membrane after fusion revealed a notable occurrence of the flip-flop process during system merging. Specifically, protonated ILs are prominently involved in the flip-flop process, whereas deprotonated ILs predominantly cluster in a disordered state within the central region of the endosomal membrane. This study underscores the pivotal role of the flip-flop process of protonated ILs in facilitating the fusion of LNDs with the endosomal membrane. Furthermore, we observed that higher cholesterol content in the endosomal membrane correlates with a lower proportion of ILs in the outer leaflet and a higher proportion in the midplane, supporting the established understanding that cholesterol interacts with other lipids to promote clustering and lateral packing. In addition, at the same pH, protonated MC3 was more prevalent in the outer leaflet compared to protonated ALC-0315, suggesting superior endosomal escape efficiency for MC3-containing LNDs. This aligns with experimental results that MC3-containing LNPs outperform ALC-0315 counterparts in RNA delivery. Lastly, we calculated the second-rank order parameter for the tails of ALC-0315 and MC3, as well as POPC tails with different ILs, to assess membrane alignment. The results showed that ALC-0315 is more disordered than MC3, and the presence of ALC-0315 slightly increases the disorder of POPC tails compared to when MC3 is present. This indicates that membranes containing ALC-0315 are more fluid and disordered. In conclusion, the molecular-level insights into ILs’ dynamics during LNDs’ endosomal escape can aid in developing effective ILs for designing optimal LNPs with high endosomal escape efficiency for RNA-based therapeutics and vaccines.

## Figures and Tables

**Figure 1 ijms-26-11960-f001:**
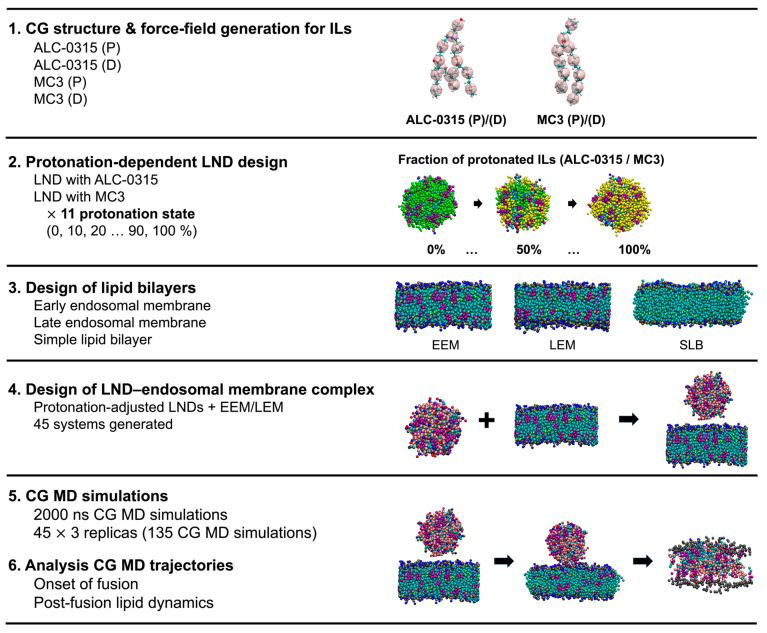
Overview of the computational workflow. Additional details of the computational steps are provided in Appendix A.

**Figure 2 ijms-26-11960-f002:**
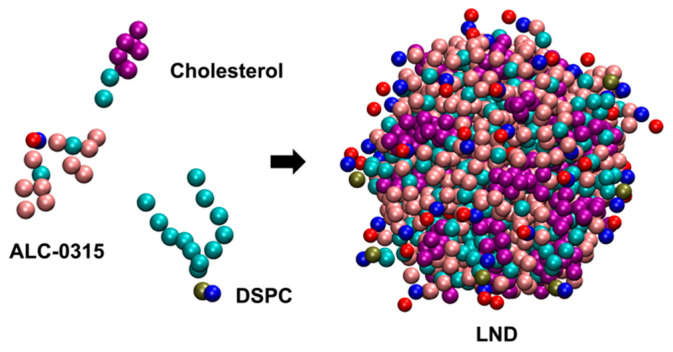
CG structure of an LND containing ALC-0315, DSPC, and cholesterol.

**Figure 3 ijms-26-11960-f003:**
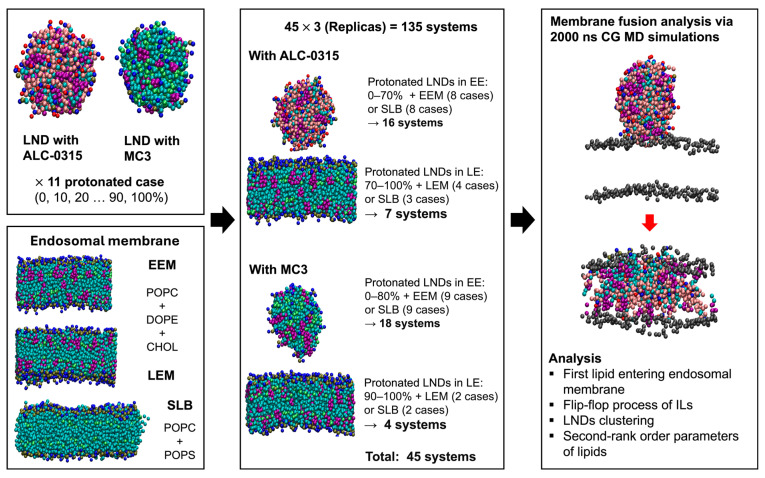
Procedures for analyzing LND and endosomal membrane fusion. From the left panel in sequence: designing LNDs, endosomal membranes, and LND–endosomal membrane complexes, followed by membrane fusion analysis using 2000 ns CG MD simulations.

**Figure 4 ijms-26-11960-f004:**
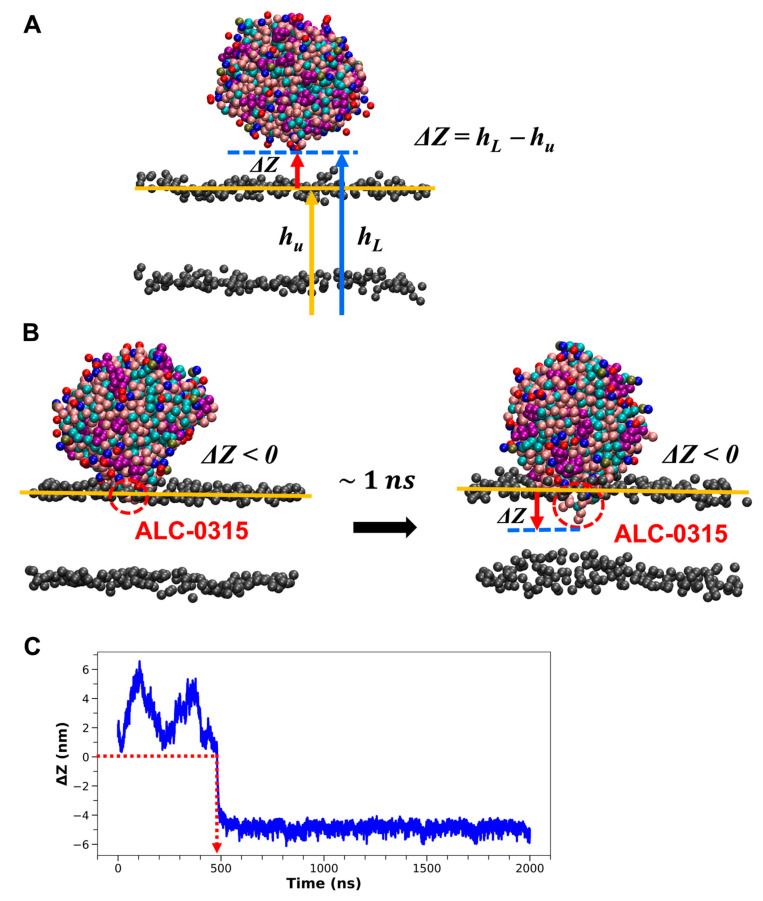
(**A**) Schematic representation for defining ΔZ. h_L_ represents the lowest z-coordinate value of the CG beads in the LND, indicated by a blue arrow. h_u_ denotes the average z-coordinate value of the upper leaflet of the lipid bilayer, shown with a yellow arrow. The blue dashed and yellow solid lines mark the z-coordinate values corresponding to h_L_ and h_u_, respectively. The PO4 CG beads of POPC or DOPE on the lipid bilayer surface are depicted in gray. ΔZ, shown with a red arrow, is calculated as h_L_ − h_u_. (**B**) Visual depiction of the onset of LND fusion with the lipid bilayer. Fusion is considered to begin when Δ*Z* shifts from positive to negative and remains negative for over 1 ns. The residue with the smallest ΔZ is identified as ALC-0315. (**C**) Time evolution of ΔZ. The red dashed arrow marks the time when ΔZ is zero.

**Figure 5 ijms-26-11960-f005:**
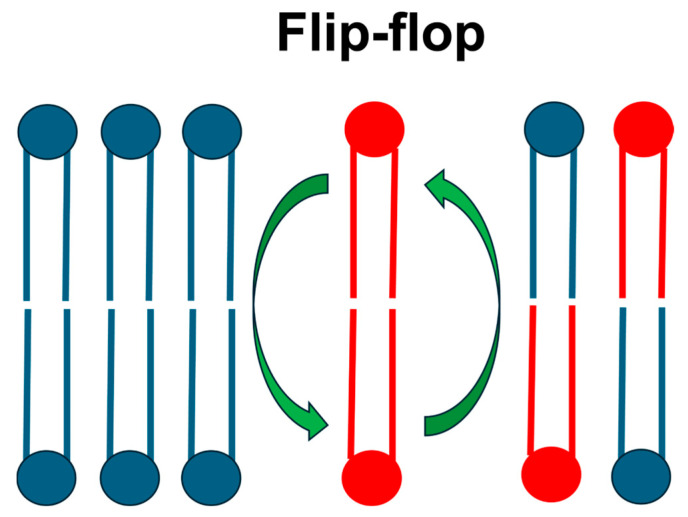
Schematic illustration of the flip-flop process in the lipid bilayer.

**Figure 6 ijms-26-11960-f006:**
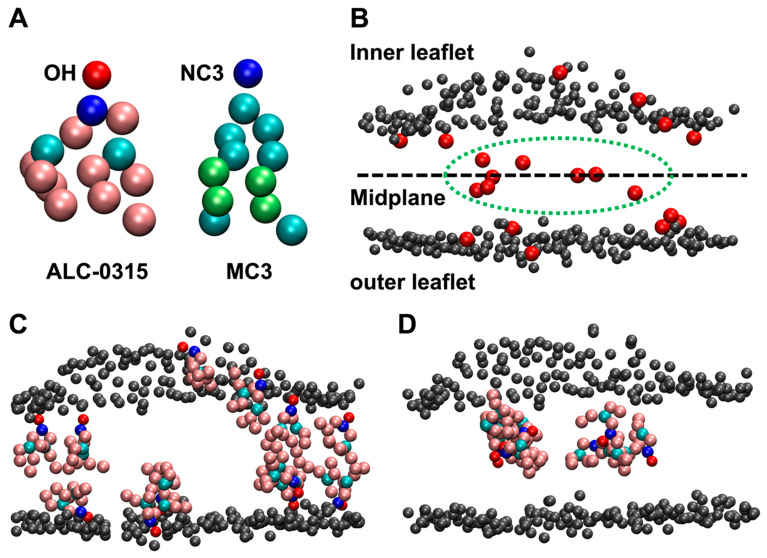
(**A**) CG representation of ALC-0315 and MC3. The OH CG bead is shown in red, while the NC3 CG bead is depicted in blue. (**B**) Classification based on the position of the OH CG bead. OH CG beads that do not belong to either the inner or outer leaflet are defined as residing in the midplane, indicated within the green dashed line. The PO4 CG beads of POPC or DOPE on the lipid bilayer surface are shown in gray. (**C**) OH CG beads of ALC-0315 that belong to the inner or outer leaflet are positioned close to the leaflet surface, with tails appearing to align nearly parallel to the bilayer normal. (**D**) ALC-0315 molecules in the midplane do not exhibit a specific orientation and appear highly disordered.

**Figure 7 ijms-26-11960-f007:**
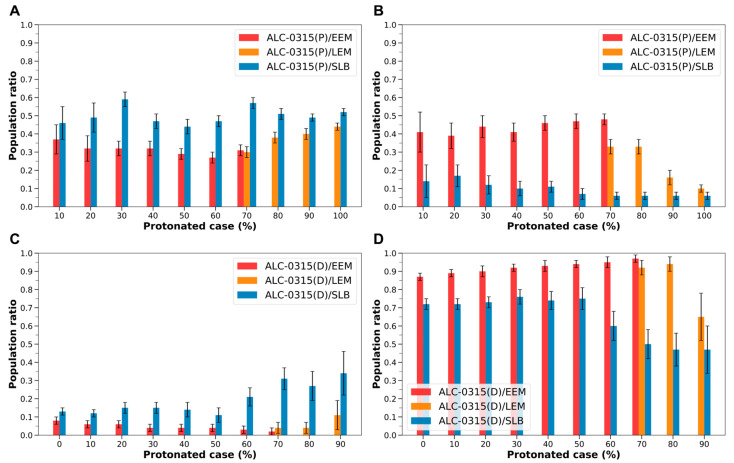
(**A**) Population ratio of protonated ALC-0315 (ALC-0315(P)) distributed in the outer leaflet during the fusion of ALC-0315-containing LNDs with EEM, LEM, and SLB, shown according to the LNDs’ protonated case. The red bars represent values for ALC-0315(P) fusing with EEM, the orange bars for LEM, and the blue bars for SLB. (**B**) Population ratio of ALC-0315(P) present in the midplane of the membrane, depicted according to the LNDs’ protonated case. (**C**) Population ratio of deprotonated ALC-0315 (ALC-0315(D)) distributed in the outer leaflet, shown according to the LNDs’ protonated case. (**D**) Population ratio of ALC-0315(D) present in the midplane of the membrane, depicted according to the LNDs’ protonated case.

**Figure 8 ijms-26-11960-f008:**
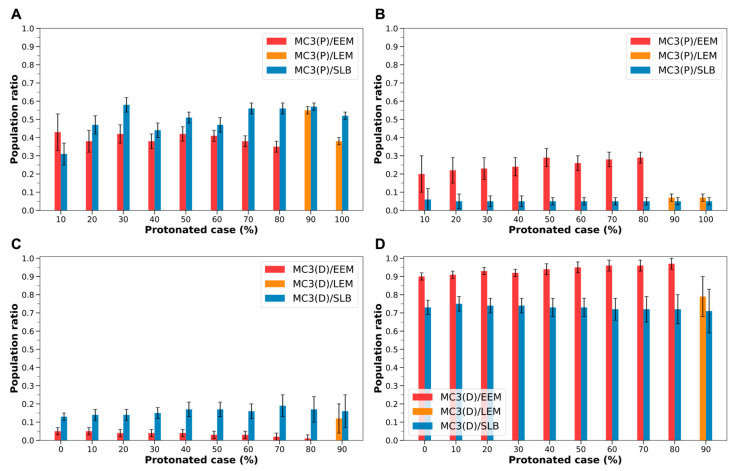
(**A**) Population ratio of protonated MC3 (MC3(P)) in the outer leaflet during the fusion of MC3-containing LNDs with EEM, LEM, and SLB, based on the LNDs’ protonated case. The red bars indicate MC3(P) fusing with EEM, the orange bars with LEM, and the blue bars with SLB. (**B**) Population ratio of MC3(P) in the midplane of the membrane, according to the LNDs’ protonated case. (**C**) Population ratio of deprotonated MC3 (MC3(D)) in the outer leaflet, according to the LNDs’ protonated case. (**D**) Population ratio of MC3(D) in the midplane of the membrane, according to the LNDs’ protonated case.

**Figure 9 ijms-26-11960-f009:**
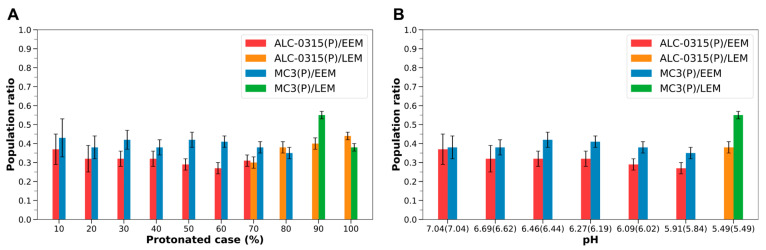
(**A**) Population ratios of ALC-0315(P) and MC3(P) present in the outer leaflet during the fusion of ALC-0315-containing and MC3-containing LNDs with EEM or LEM, based on the LNDs’ protonated case. Red and orange bars indicate the population ratio for ALC-0315(P) when ALC-0315-containing LNDs fuse with EEM and LEM, respectively. Blue and green bars depict the ratio for MC3(P) when MC3-containing LNDs fuse with EEM and LEM, respectively. (**B**) Population ratios of ALC-0315(P) and MC3(P) present in the outer leaflet, shown according to the pH of the LNDs. pH values outside the parentheses correspond to ALC-0315-containing LNDs, whereas pH values inside the parentheses correspond to MC3-containing LNDs.

**Figure 10 ijms-26-11960-f010:**
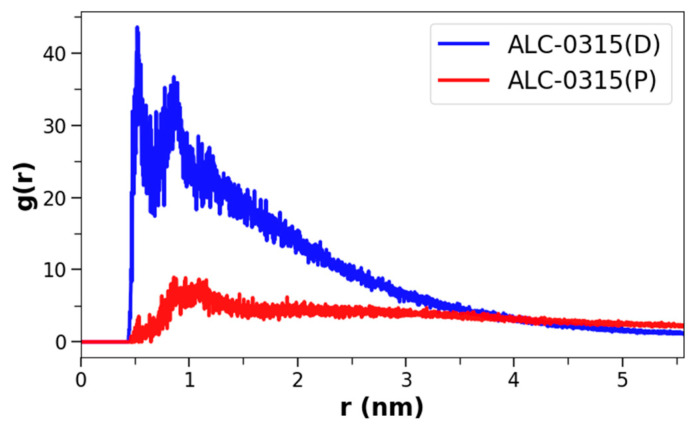
RDFs, g(r) of NC3 CG beads for ALC-0315(P) and ALC-0315(D) after fusion of LNDs with 50% protonated ALC-0315 and EEM. Here, *r* represents the distance between two beads. ALC-0315(P) is shown in red, and ALC-0315(D) in blue.

**Figure 11 ijms-26-11960-f011:**
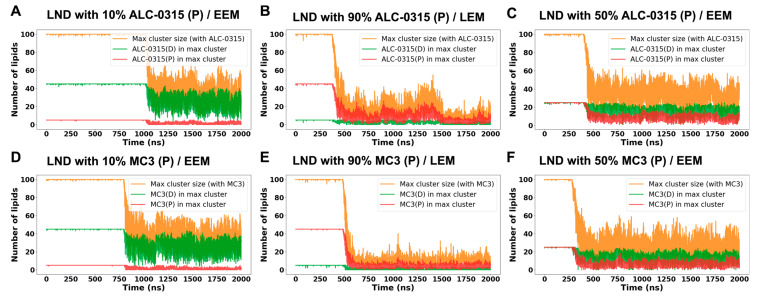
Time evolution of the maximum cluster size of LNDs containing ILs and the number of protonated and deprotonated molecules within the max cluster during fusion with EEM or LEM. (**A**) LNDs containing 10% ALC-0315(P) fusing with EEM. (**B**) LNDs containing 90% ALC-0315(P) fusing with LEM. (**C**) LNDs containing 50% ALC-0315(P) fusing with EEM. (**D**) LNDs containing 10% MC3(P) fusing with EEM. (**E**) LNDs containing 90% MC3(P) fusing with LEM. (**F**) LNDs containing 50% MC3(P) fusing with EEM. Orange lines indicate the size of the largest cluster containing the respective IL, green lines show the number of deprotonated molecules [ALC-0315(D) or MC3(D)], and red lines represent the number of protonated molecules [ALC-0315(P) or MC3(P)] within the max cluster.

**Figure 12 ijms-26-11960-f012:**
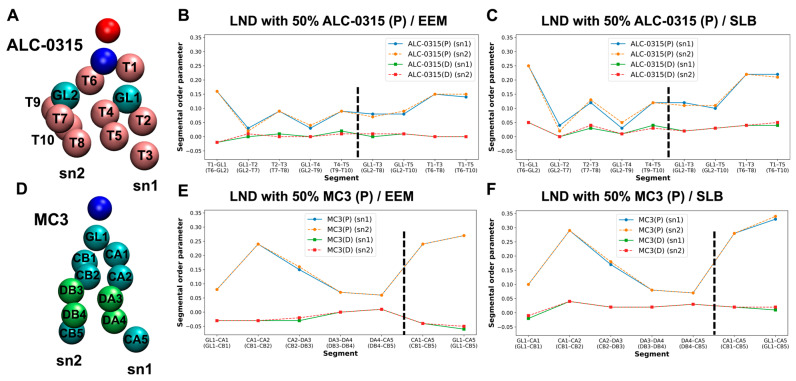
(**A**) Bead types along the tails of ALC-0315. (**B**,**C**) Segmental order parameters of ALC-0315 tails after fusion of LNDs containing 50% ALC-0315(P) with EEM (**B**) and SLB (**C**). The left and right sides of the black dashed line represent bond order parameters for consecutive tail bonds and non-consecutive bead vectors, respectively. Blue and orange indicate the sn1 and sn2 tails of ALC-0315(P), while green and red represent the sn1 and sn2 tails of ALC-0315(D). (**D**) Bead types along the tails of MC3. (**E**,**F**) Segmental order parameters of MC3 tails after fusion of LNDs containing 50% MC3(P) with EEM (**E**) and SLB (**F**). The color scheme follows that in (**B**): blue/orange for MC3(P), and green/red for MC3(D).

**Figure 13 ijms-26-11960-f013:**
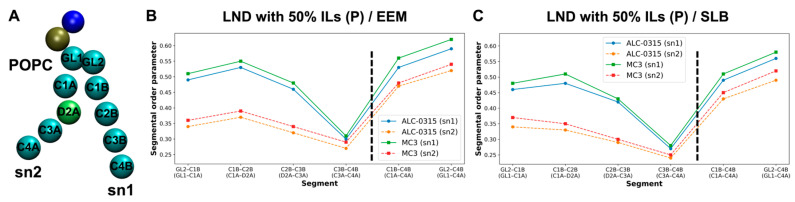
(**A**) Bead types along the tails of POPC. (**B**,**C**) Segmental order parameters of POPC tails after fusion of LNDs containing 50% ILs(P) with EEM (**B**) or SLB (**C**). The left and right sides of the black dashed line represent bond order parameters for consecutive tail bonds and non-consecutive bead vectors, respectively. Blue and orange indicate the sn1 and sn2 tails of POPC when the IL is ALC-0315, while green and red represent the sn1 and sn2 tails of POPC when the IL is MC3.

**Table 1 ijms-26-11960-t001:** Ratios of protonated and deprotonated ALC-0315 in the outer leaflet and midplane during the fusion of ALC-0315-containing LNDs with EEM or LEM, based on the protonated case (%). Here, (P) denotes the protonated form and (D) denotes the deprotonated form.

Protonated Case (%)	Outer Leaflet Ratio (P)	Outer Leaflet Ratio (D)	Midplane Ratio (P)	Midplane Ratio (D)
0	–	0.08 ± 0.02	–	0.87 ± 0.02
10	0.37 ± 0.08	0.06 ± 0.02	0.41 ± 0.11	0.89 ± 0.02
20	0.32 ± 0.07	0.06 ± 0.02	0.39 ± 0.07	0.90 ± 0.03
30	0.32 ± 0.04	0.04 ± 0.02	0.44 ± 0.06	0.92 ± 0.02
40	0.32 ± 0.04	0.04 ± 0.02	0.41 ± 0.05	0.93 ± 0.03
50	0.29 ± 0.03	0.04 ± 0.02	0.46 ± 0.04	0.94 ± 0.02
60	0.27 ± 0.03	0.03 ± 0.02	0.47 ± 0.04	0.95 ± 0.03
70	0.31 ± 0.03/ 0.30 ± 0.03	0.02 ± 0.02/ 0.04 ± 0.03	0.48 ± 0.03/ 0.33 ± 0.04	0.97 ± 0.02/ 0.92 ± 0.04
80	0.38 ± 0.03	0.04 ± 0.03	0.33 ± 0.04	0.94 ± 0.04
90	0.40 ± 0.03	0.11 ± 0.08	0.16 ± 0.04	0.65 ± 0.13
100	0.44 ± 0.02	–	0.10 ± 0.02	–

**Table 2 ijms-26-11960-t002:** Outer leaflet and midplane ratios of protonated and deprotonated ALC-0315 during the fusion of ALC-0315-containing LNDs with SLB, based on the protonated case (%).

Protonated Case (%)	Outer Leaflet Ratio (P)	Midplane Ratio (P)	Outer Leaflet Ratio (D)	Midplane Ratio (D)
0	–	–	0.13 ± 0.02	0.72 ± 0.03
10	0.46 ± 0.09	0.14 ± 0.09	0.12 ± 0.02	0.72 ± 0.03
20	0.49 ± 0.08	0.17 ± 0.06	0.15 ± 0.03	0.73 ± 0.03
30	0.59 ± 0.04	0.12 ± 0.05	0.15 ± 0.03	0.76 ± 0.04
40	0.47 ± 0.04	0.10 ± 0.04	0.14 ± 0.04	0.74 ± 0.05
50	0.44 ± 0.04	0.11 ± 0.03	0.11 ± 0.04	0.75 ± 0.06
60	0.47 ± 0.03	0.07 ± 0.03	0.21 ± 0.05	0.60 ± 0.08
70	0.57 ± 0.03	0.06 ± 0.02	0.31 ± 0.06	0.50 ± 0.08
80	0.51 ± 0.03	0.06 ± 0.02	0.27 ± 0.08	0.47 ± 0.09
90	0.49 ± 0.02	0.06 ± 0.02	0.34 ± 0.12	0.47 ± 0.13
100	0.52 ± 0.02	0.06 ± 0.02	–	–

**Table 3 ijms-26-11960-t003:** pH values for each protonated case of ALC-0315-containing and MC3-containing LNDs.

Protonated Case (%)	pH of ALC-0315-Containing LNDs	pH of MC3-Containing LNDs
0	–	–
10	7.04	7.39
20	6.69	7.04
30	6.46	6.81
40	6.27	6.62
50	6.09	6.44
60	5.91	6.19
70	5.72	6.02
80	5.49	5.84
90	5.14	5.49
100	–	–

## Data Availability

The initial structures for LNDs containing ALC-0315 and MC3, the endosomal membrane, and the LND–endosomal membrane complex used in CG MD simulations, along with the required topology files, input files, and scripts for data analysis, are available to the public on GitHub (https://github.com/mogam-ai/LNP_CG-MD (accessed on 18 November 2025)).

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
