# Peer review of "Coarse-Grained Molecular Dynamics Simulations of Lipid Nanodroplets and Endosomal Membranes: Focusing on the Fusion Mechanisms"

_ijms, 2025, doi:10.3390/ijms262411960_

Round 1
Reviewer 1 Report
Comments and Suggestions for Authors
Reviewer Report
Manuscript ID: IJMS-3960198
Title: Coarse-grained molecular dynamics simulations of lipid nanodroplets and endosomal membranes: Focusing on the fusion mechanism
This manuscript presents a comprehensive coarse-grained molecular dynamics (CG-MD) study addressing the fusion mechanism between lipid nanodroplets (LNDs) containing ionizable lipids (ILs) and endosomal membranes under varying pH and compositional environments. The topic is timely and relevant to the growing interest in understanding endosomal escape processes of lipid nanoparticles (LNPs) used for RNA delivery.
The work is conceptually well-motivated and logically organized. The authors have systematically designed multiple LND–membrane systems, incorporated both ALC-0315 and MC3 lipids, and analyzed flip-flop dynamics, clustering, and membrane ordering. The major conclusions—that protonated ILs initiate fusion and undergo flip-flop more efficiently, and that MC3-containing systems show higher endosomal escape potential—are physically meaningful and consistent with experimental observations in the literature.
Nevertheless, several aspects require clarification or further discussion before the manuscript can be considered for publication.
- The study builds on established Martini-based CG-MD methods for membrane fusion, yet its strength lies in applying the framework specifically to ionizable lipids relevant to therapeutic LNPs. While the manuscript is comprehensive, the novelty could be more clearly articulated. For instance, how does this work advance beyond prior CG studies such as those by Bruininks et al. (eLife, 2020) or ÄŒechová et al. ( Biomol. Struct. Dyn., 2024)? A sharper statement on the unique mechanistic insight—such as the role of protonated lipid flip-flop as a quantifiable kinetic descriptor—would strengthen the paper’s scientific contribution.
- The authors developed CG parameters for ALC-0315 and MC3 using Martini 3, guided by AA simulations. Although qualitative comparisons (bond and angle distributions, SASA) are shown, more quantitative validation metrics—e.g., density, radius of gyration, or radial distribution deviation between AA and CG—would improve confidence in the parameter transferability. Also it would make it more complete and also increase reproducability of the study, when they provide more details on the CG-FF parameterization instead of just figure legends.
- Each trajectory spans 2 μs with three replicas, which is commendable. However, fusion and flip-flop events are stochastic processes that may depend on initial configurations. The authors should discuss convergence across replicas and clarify how representative these trajectories are. Presenting time-evolution plots for fusion onset or IL migration fraction would support the robustness of conclusions.
- The “simple lipid bilayer” (80% POPC + 20% POPS) is a pragmatic model but lacks components such as BMP or sphingomyelin known to influence endosomal curvature and charge asymmetry. The authors are encouraged to discuss how these simplifications might affect the observed lipid dynamics and the generality of the mechanistic conclusions.
- The protonation states are fixed according to the Henderson–Hasselbalch equation. In reality, IL protonation is dynamic and can vary locally near the membrane interface. The authors should acknowledge this limitation and comment on whether a constant-pH or adaptive protonation model could capture additional mechanistic details in future studies.
- The manuscript refers to simulation input and analysis files available on GitHub. However, the link provided appears inactive at this time, preventing independent verification. The authors should ensure that supporting data and scripts are properly accessible for reproducibility once the paper is published as well as under review.
Summary and Recommendation
This is a technically solid and scientifically relevant manuscript that offers mechanistic insights into ionizable lipid behavior during LNP–membrane fusion, a process of major importance for mRNA therapeutics. The methodology is appropriate, and the conclusions are supported by the presented analyses.
However, several methodological clarifications and quantitative validations are necessary to improve the rigor and reproducibility of the study.
Recommendation: Major Revision
Author Response
- The study builds on established Martini-based CG-MD methods for membrane fusion, yet its strength lies in applying the framework specifically to ionizable lipids relevant to therapeutic LNPs. While the manuscript is comprehensive, the noveltycould be more clearly articulated. For instance, how does this work advance beyond prior CG studies such as those by Bruininks et al.(eLife, 2020) or ÄŒechová et al. ( Biomol. Struct. Dyn., 2024)? A sharper statement on the unique mechanistic insight—such as the role of protonated lipid flip-flop as a quantifiable kinetic descriptor—would strengthen the paper’s scientific contribution.
Author reply: We sincerely thank the reviewer for recognizing the value of our work and for providing valuable suggestions to further enhance the paper’s scientific contribution. As advised, we have strengthened the articulation of our study’s novelty compared with prior CG studies. In particular, we now more clearly emphasize that our CG MD simulation system incorporates the CG force field parameters for ionizable lipids that we newly developed specifically for therapeutic LNPs, and that we used these parameters to design our LNP models. We also highlight our key finding, namely the mechanistic insights we uncovered regarding LND and endosomal membrane fusion, including the unique role and characteristics of protonated ILs in this process. These points have been more clearly and strongly emphasized in the Introduction. We once again sincerely appreciate the reviewer’s insightful comments.
Changed on p. 4: Thus, we performed hundreds of microseconds of CG MD simulations using Martini 3 force fields [27, 29] and IL parameters developed with the Martini 3 framework to investigate the fusion process of LNPs into the endosomal membrane.
→ Thus, we performed hundreds of microseconds of CG MD simulations using Martini 3 force fields [27, 29] together with newly developed Martini 3–compatible CG parameters for ILs to investigate the fusion process of LNPs into the endosomal membrane.
Added on p. 4: However, previous computational studies did not clarify how protonated and deprotonated ILs differentially contribute to membrane fusion, leaving the roles of ILs that depend on their protonation state largely unresolved [23, 24]. In contrast, our study shows that protonation dependent IL behavior, especially the flip-flop of protonated ILs, is a key mechanistic factor that contributes to the fusion required for endosomal escape.
- The authors developed CG parameters for ALC-0315 and MC3 using Martini 3, guided by AA simulations. Although qualitative comparisons (bond and angle distributions, SASA) are shown, more quantitative validation metrics—e.g., density, radius of gyration, or radial distribution deviation between AA and CG—would improve confidence in the parameter transferability. Also it would make it more complete and also increase reproducability of the study, when they provide more details on the CG-FF parameterization instead of just figure legends.
Author reply: We sincerely thank the reviewer for this thoughtful and constructive suggestion. In response, we have strengthened the quantitative validation of our CG parameters by comparing mass density, radius of gyration, and radial distribution deviations between AA and CG simulations for both protonated and deprotonated ALC-0315 and MC3. The corresponding results, now provided in Tables S7–S9, show that the CG models reproduce AA structural properties within reasonable MARTINI accuracy ranges (density deviations ≤2.7%, Rg deviations ≤0.05 nm, and RMS Δg ≈ 0.20). To further enhance transparency and reproducibility, we have added a dedicated section to the Supporting Information (“CG force field parameterization for ALC-0315 and MC3”), which describes the AA-to-CG mapping, extraction of bonded parameters, and the iterative refinement procedure. A brief description of these additions has also been incorporated into the main text. We hope these revisions address the reviewer’s concern and improve the clarity and completeness of our work.
Added on p. 32: Detailed procedures for obtaining the initial CG structures and CG FF parameters are provided in the Supporting Information under the section “CG force field parameterization for ALC-0315 and MC3.” To strengthen the reliability and transferability of the newly developed CG parameters, we additionally performed a quantitative validation of the CG models against AA reference simulations. Specifically, we compared mass density, radius of gyration, and the root mean square deviation of RDFs (RMS Δg) between AA and CG simulations for both protonated and deprotonated ALC-0315 and MC3. All metrics showed AA–CG agreement with deviations within reasonable CG–AA accuracy ranges, and the detailed numerical values are provided in Tables S7–S9.
- Each trajectory spans 2 μs with three replicas, which is commendable. However, fusion and flip-flop events are stochastic processes that may depend on initial configurations. The authors should discuss convergence across replicasand clarify how representative these trajectories are. Presenting time-evolution plots for fusion onset or IL migration fraction would support the robustness of conclusions.
Author reply: We sincerely thank the reviewer for this constructive comment. As noted, fusion is a stochastic process, and therefore the fusion onset times naturally vary among the three replicas for each system. Nevertheless, when examining the IL migration fractions (outer leaflet and membrane midplane) as a function of time, we find that all replicas exhibit similar behavior in the post-fusion regime. In particular, during the 1500–2000 ns window—where all systems reside in the post-fusion regime—the IL distributions across the three replicas fall within similar population ratio ranges despite the expected fluctuations. Importantly, all replicas consistently show that protonated ILs more frequently populate the outer leaflet, whereas deprotonated ILs preferentially accumulate in the membrane midplane, in agreement with our original conclusions. Following the reviewer’s suggestion, we have added time-evolution plots for both fusion onset (ΔZ) and IL migration fractions to the Supporting Information (Figure S2). To keep the presentation concise, we present one representative example—a 50%-protonated ALC-0315 LND fused with EEM—while the full IL migration data from all 45 systems and their three independent replicas are reported in Tables S1–S4. The corresponding sections of the manuscript have been updated accordingly.
Added on p. 15-16: To further assess the robustness of our simulations, we examined the migration rates obtained from the three replicas and confirmed that they exhibit consistent trends across systems (Tables S1–S4). In addition, time-evolution plots of fusion onset (ΔZ) and IL migration fractions for a representative system are provided in Figure S2, illustrating the temporal behavior leading to the post-fusion regime.
Changed on p. 16: These systems …
→ The 45 systems …
- The “simple lipid bilayer” (80% POPC + 20% POPS) is a pragmatic model but lacks components such as BMP or sphingomyelin known to influence endosomal curvature and charge asymmetry. The authors are encouraged to discuss how these simplifications might affect the observed lipid dynamics and the generality of the mechanistic conclusions.
Author reply: We thank the reviewer for the insightful comment. As noted by the reviewer, our endosomal membrane models did not include sphingomyelin (SM) or BMP. In response to this suggestion, we have added a detailed discussion in Section 2.4 of the Results and Discussion to address how the inclusion of SM and BMP would influence our findings.
Added on p. 19-20: Although our endosomal models do not explicitly include sphingomyelin (SM) or bis(monoacylglycerol)phosphate (BMP), incorporating these lipids would likely reinforce the trends observed in our simulations across SLB, EEM, and LEM. SM is relatively enriched in EEM, where its strong interaction with cholesterol promotes lateral packing and reduces membrane fluidity [50, 51]. In contrast, BMP is uniquely localized to the inner leaflet of late endosomes and lysosomes and is known to enhance the activity of lipid transfer proteins such as saposins and Niemann-Pick disease type C2, thereby facilitating cholesterol mobilization and extraction [50, 51]. These properties are consistent with the lower cholesterol content and higher membrane fluidity observed in LEM. Therefore, even when SM and BMP are considered, the simulation-derived conclusion that reduced cholesterol content increases IL flip-flop would remain unchanged. Furthermore, BMP is a negatively charged, cone-shaped phospholipid restricted to the inner leaflet of late endosomes [52]. Owing to these structural and electrostatic features, BMP-rich membranes are thought to facilitate membrane fusion [53]. Consequently, in BMP-rich and highly protonated late endosomal environments, the flip-flop of protonated ILs would be expected to increase.
- The protonation states are fixed according to the Henderson–Hasselbalch equation. In reality, IL protonation is dynamic and can vary locally near the membrane interface. The authors should acknowledge this limitation and comment on whether a constant-pH or adaptive protonation modelcould capture additional mechanistic details in future studies.
Author reply: We sincerely appreciate the reviewer’s insightful comment regarding the limitation associated with using fixed protonation states. As noted by the reviewer, ionizable lipids can undergo protonation–deprotonation transitions driven by local pH and electrostatic variations near membrane interfaces. Although such dynamic effects cannot be captured within the current coarse-grained framework, we have now clarified this limitation in the revised manuscript. We further note that the use of constant-pH MD simulations or other adaptive protonation approaches in future work may better represent environment-dependent protonation dynamics and provide additional mechanistic insight.
Added on p. 9-10: Although these simulations provided meaningful results, the conventional CG MD approach used in this study employs fixed protonation states for ILs. Consequently, it cannot capture the protonation changes that may occur due to the local lipid environment [39], such as during the fusion of LNDs with endosomal membranes. To address this limitation, constant-pH MD simulations or other adaptive protonation approaches, which allow protonation states to respond to the surrounding environment, could offer a more accurate description of lipid dynamics during the fusion process [40, 41]. Despite their high computational cost and the challenges associated with achieving convergence in lipid systems with pH-dependent structural transitions and force field sensitivity [42], such methods may reveal additional mechanistic details in future studies.
- The manuscript refers to simulation input and analysis files available on GitHub. However, the link provided appears inactive at this time, preventing independent verification. The authors should ensure that supporting data and scripts are properly accessible for reproducibility once the paper is published as well as under review.
Author reply: We thank the reviewer for pointing this out. We confirmed that the GitHub repository had restricted visibility, which prevented access during the review process. The repository settings have now been corrected, and the link has been reactivated to ensure that all simulation input files and analysis scripts are publicly accessible. We appreciate the reviewer’s careful attention to this matter.
Changed on p. 36: https://github.com/mogam-ai/LNP_CG-MD (accessed on 15 October 2025)
→ https://github.com/mogam-ai/LNP_CG-MD (accessed on 18 November 2025)
Reviewer 2 Report
Comments and Suggestions for Authors
The paper is hard to read. There are many conditions and compositions that are difficult to follow. For the sake of clarity, I would advise a route sheet to guide the reader, specifically to those that are no familiarized with computational methods in order to gain knowledge in relation to real systems,s
Author Response
The paper is hard to read. There are many conditions and compositions that are difficult to follow. For the sake of clarity, I would advise a route sheet to guide the reader, specifically to those that are no familiarized with computational methods in order to gain knowledge in relation to real systems.
Author reply: Thank you for this helpful suggestion. We agree that the number of systems and simulation conditions may make the workflow difficult to follow, especially for readers who are less familiar with computational methods. To improve clarity, we have added a new schematic overview summarizing the overall computational workflow (now presented as Figure 1 in the Introduction). This route sheet visually outlines the major steps of our approach, including IL parameter preparation, protonation-dependent LND design, lipid-bilayer construction, LND–membrane complex assembly, CG MD simulations, and fusion analysis. In addition, a fully detailed version of the computational procedure is now provided in Figure S1 in the Supplementary Information, where each step is described more specifically. We believe that the combination of Figure 1 (overview) and Figure S1 (detailed workflow) greatly enhances the readability and accessibility of our methodology.
Reviewer 3 Report
Comments and Suggestions for Authors
Deepen the biological and experimental anchoring of the simulation results.
Provide explicit comparisons with published quantitative data (endosomal escape efficiencies, fusion rates, membrane perturbation energies). This will greatly increase the manuscript’s impact.
Standardize units, abbreviations, and numerical formats.
Verify reference formatting according to IJMS standards
Author Response
Deepen the biological and experimental anchoring of the simulation results.
Author reply: We sincerely thank the reviewer for this valuable and insightful suggestion. In response, we have strengthened the biological and experimental grounding of our simulation results. Specifically, we have linked one of our key findings—namely, that deprotonated ILs tend to cluster while protonated ILs remain more dispersed, and that deprotonated ILs predominantly occupy the membrane midplane whereas protonated ILs localize near the surface—to previously reported experimental observations. We believe that incorporating these studies substantially enhances the robustness and biological relevance of our conclusions.
Added on p. 25-26: Our simulation results are consistent with earlier studies using cryo-TEM, SAXS, NMR, and MD simulations, which reported that neutral ionizable aminolipids reside between the two membrane leaflets, while protonated lipids remain at the bilayer surface [58, 59]. Furthermore, cryo-TEM analyses of Lipid-5–containing LNPs have revealed a single lamellar region surrounding an amorphous core [60], and MD simulations have captured the formation of amorphous droplets composed of neutral Lipid-5 within the hydrophobic core [61]. Together, these experimental and computational observations support our finding that deprotonated ILs preferentially cluster in the hydrophobic midplane region of the membrane.
Provide explicit comparisons with published quantitative data (endosomal escape efficiencies, fusion rates, membrane perturbation energies). This will greatly increase the manuscript’s impact.
Author reply: We thank the reviewer for this valuable suggestion, which we believe significantly strengthens the overall impact of the manuscript. In accordance with the reviewer’s comment, we have incorporated published quantitative data on endosomal escape efficiencies. Specifically, we now explicitly compare MC3- and ALC-0315–containing LNPs by citing experimental measurements showing that MC3 achieves 5% cytosolic delivery, whereas ALC-0315 achieves 4%, supporting our simulation results. This information has been added to the revised manuscript.
Regarding the reviewer’s request for quantitative comparisons of fusion rates and membrane perturbation energies, we agree that these parameters would further enrich the mechanistic discussion. However, to the best of our knowledge, accurate quantitative measurements for fusion rates or membrane perturbation energies specific to MC3 or ALC-0315 have not yet been reported. We fully acknowledge the importance of these parameters, and we have added a statement in the Results and discussion section indicating that quantifying fusion kinetics and membrane perturbation energetics represents an important direction for future work. We believe that this revision appropriately reflects the reviewer’s insightful recommendation while maintaining accuracy with respect to the current experimental literature.
Changed on p. 23: This aligns with experimental results showing that MC3-containing LNPs have higher endosomal escape efficiency than ALC-0315–containing LNPs in intravenous delivery of mRNA-based drugs, where tissue clearance properties are less critical.
→ This aligns with experimental results showing that MC3-containing LNPs have higher endosomal escape efficiency than ALC-0315–containing LNPs in mRNA delivery studies, with MC3 achieving 5% cytosolic delivery compared to 4% for ALC-0315 [54]. Although the numerical difference is modest, endosomal escape is widely regarded as a major bottleneck in intracellular drug delivery, with typically fewer than 5% of nucleic acid cargo reaching the cytosol. Therefore, even a 1% difference can represent a biologically meaningful distinction. While our current study does not address membrane perturbation energy directly, future computational studies that map the free-energy landscape of lipid insertion or quantitatively estimate fusion rates could provide a deeper mechanistic understanding of LNP–membrane interactions and endosomal escape.
Standardize units, abbreviations, and numerical formats.
Author reply: Thank you for the helpful suggestion. We have carefully reviewed the manuscript to ensure consistent use of units, abbreviations, and numerical formatting. During this revision, we corrected minor typographical issues (e.g., replacing hyphens with en dashes for numerical ranges and applying SI-compliant units such as nm³), fixed several misspellings of SLB, and added the full name of POPC at its first occurrence. We also standardized abbreviations throughout the text, including defining (P) and (D) in the caption of Table 1 for clarity. We appreciate the reviewer’s comment, which helped improve the overall consistency of the manuscript.
Verify reference formatting according to IJMS standards
Author reply: We appreciate this reminder. All references have been thoroughly checked and reformatted to comply with the IJMS reference style, including journal abbreviations, page formatting, and removal of issue numbers. We have also ensured full consistency between the in-text citations and the reference list.
Round 2
Reviewer 2 Report
Comments and Suggestions for Authors
The paper is greatly improved although iy is addressed to MD specialist. however, I think it may be published in the present form.